# Bismuth-Germanate Glasses: Synthesis, Structure, Luminescence, and Crystallization

Ksenia Serkina [1], Irina Stepanova [1], Aleksandr Pynenkov [2], Maria Uslamina [2], Konstantin Nishchev [2], Kirill Boldyrev [3], Roman Avetisov [1] and Igor Avetissov [1,*]

[1] Department of Chemistry and Technology of Crystals, D. Mendeleev University of Chemical Technology of Russia (MUCTR), 125480 Moscow, Russia; serkina.k.v@muctr.ru (K.S.); stepanova.i.v@muctr.ru (I.S.); armoled@mail.ru (R.A.)

[2] Institute of High Technologies and New Materials, National Research Mordovia State University, 430005 Saransk, Russia; alekspyn@yandex.ru (A.P.); uslaminam@mail.ru (M.U.); nishchev@inbox.ru (K.N.)

[3] Laboratory of Fourier-Spectroscopy, Institute for Spectroscopy RAS, 108840 Troitsk, Russia; kn.boldyrev@gmail.com

* Correspondence: avetisov.i.k@muctr.ru

**Abstract:** Bismuth-germanate glasses, which are well known as a promising active medium for broadband near-infrared spectral range fiber lasers and as an initial matrix for nonlinear optical glass ceramics, have been synthesized in a 5–50 mol% $Bi_2O_3$ wide concentration range. Their structural and physical characteristics were studied by Raman and FT-IR spectroscopy, differential scanning calorimetry, X-ray diffraction, optical, and luminescence methods. It has been found that the main structural units of glasses are $[BiO_6]$ and $[GeO_4]$. The growth in bismuth oxide content resulted in an increase in density and refractive index. The spectral and luminescent properties of glasses strongly depended on the amount of bismuth active centers. The maximum intensity of IR luminescence has been achieved for the $5Bi_2O_3$-$95GeO_2$ sample. The heat treatment of glasses resulted in the formation of several crystalline phases, the structure and amount of which depended on the initial glass composition. The main phases were non-linear $Bi_2GeO_5$ and scintillating $Bi_4Ge_3O_{12}$. Comparing with the previous papers dealing with bismuth and germanium oxide-based glasses, we enlarge the range of $Bi_2O_3$ concentration up to 50 mol% and decrease the synthesis temperature from 1300 to 1100 °C.

**Keywords:** bismuth-germanate glass; bismuth active centers; IR-luminescence; crystallization

## 1. Introduction

The $Bi_2O_3$-$GeO_2$ system has a wide glass transition region up to 85.7 mol% $Bi_2O_3$ [1,2]. The basis of the structural network of bismuth-germanate glasses is $[GeO_4]^{4-}$-tetrahedra [3]. Bismuth oxide, as a modifier, creates additional bonds in glass, strengthening it. It is generally accepted that $Bi^{3+}$ ions are predominantly in octahedral coordination in glass, but it can be varied from octahedral $[BiO_6]^{6-}$ to pyramidal $[BiO_3]^{3-}$ at Bi-concentration growth. Along with $Bi^{3+}$, bismuth can exist in other charge states in glasses [4].

Bismuth-containing glasses have a high refractive index and high density; they are transparent in the visible and IR spectral ranges [5,6]. Increased researchers' attention to these glasses arose after the discovery of a unique luminescence in the 1100–1500 nm range, the source of which is bismuth active centers (BACs) [7]. The structure of these centers has been subjected to changes over the past 20 years [8–10]. Up to date, the prevailing opinion is that BAC has a complex active structure, which is a combination of bismuth ions in low oxidation states and oxygen vacancies [11]. Understanding the nature of these centers would make it possible to optimize laser active media for the near-IR range.

In addition to active optics, bismuth-germanate glasses are used to produce glass-ceramic materials since the glass formation region of the $Bi_2O_3$-$GeO_2$ system includes

the compositions of different crystalline phases: "metastable" $Bi_2GeO_5$ with ferroelectric characteristics [12], and $Bi_4Ge_3O_{12}$ with scintillation properties [13].

All this makes bismuth-germanate glasses both unique and multipurpose materials. Thus, the goal of the present research was to investigate the properties of bismuth-germanate glasses synthesized in a wide range of bismuth and germanium oxide concentrations for further application in various fields of science and technology.

## 2. Materials and Methods

We synthesized $xBi_2O_3(100-x)GeO_2$ glasses in the 5–50 mol% $Bi_2O_3$ concentration range with a 5 mol% step. We used $Bi_2O_3$ 99.999 wt% and $GeO_2$ 99.995 wt% purchased from LANHIT LTD (Russia, Moscow). For a better presentation of the results, the samples were signed as x(100-x). For example, the $15Bi_2O_3$-$85GeO_2$ composition was signed as 15-85 (Sample ID). Glasses were synthesized in corundum crucibles at 1100 °C for 30 min by the standard melt-quenching technique with casting onto a metal substrate at room temperature. Thermal stresses were removed at 350 °C for 3 h, followed by cooling at a rate of ~50 °C/h. Polished parallel plates with 2 mm thickness were made from glasses for further studies.

The elemental analysis of the synthesized glasses was carried out using an X-ray spectral energy-dispersive microanalyzer (EDS Oxford Instruments X-MAX-50) on the base of a Tescan VEGA3-LMU scanning electron microscope (TESCAN ORSAY HOLDING, Brno, Czech Republic). Raman spectra were recorded on a QE65000 spectrophotometer (Ocean Optics, Largo, FL, USA) using a 785 nm excitation laser in the frequency shift range of 200–2000 $cm^{-1}$ in backscattering geometry. IR transmission spectra were recorded on a Tensor 27 IR-Fourier spectrometer (Bruker, Ettlingen, Germany) in the 400–8000 $cm^{-1}$ range.

The characteristic temperatures of the samples were determined by different scanning calorimetry methods using a DSC 404 F1 Pegasus instrument (Erich Netzsch GmbH & Co. Holding KG, Selb, Germany). The measurements were carried out for 100–120 mg samples placed in platinum crucibles at a 20 mL/min airflow and 10 °C/min heating rate. The density of the samples was determined by the hydrostatic method using a M-ER123 ACF JR-150.005 TFT balance (Mercury WP Tech Group Co., Ltd., Incheon, Republic of Korea) with an accuracy of 0.005 $g/cm^3$. The refractive index ($n_D > 1.78$) was determined using a MIN-8 optical microscope (LOMO JSC, Saint Peterburg, Russia) by measuring the shift of the refracted beam at different preset tilt angles of sample plates located on a special stage.

The absorption spectra were recorded on a UNICO 2800 (UV/VIS) spectrophotometer (United Products & Instruments, Suite E Dayton, NJ, USA) in the 190–1100 nm wavelength range with a 1 nm step. The luminescence spectra were recorded on an IFS 125HR FT-IR spectrometer (Bruker, Ettlingen, Germany) using an original self-made luminescent module. Luminescence was excited by diode lasers (CNI, Changchun, China) with wavelengths of 405, 425, 525, 650, and 805 nm; the power density on the sample was 100 $mW/mm^2$, and the spectral resolution was 12 $cm^{-1}$.

The original glasses were subjected to heat treatment for 2 h at various temperatures based on the DSC data. The structure of the crystalline phases was determined by X-ray diffraction using an Equinox-2000 diffractometer (Inel SAS, Artenay, France) with a linear CCD detector with a step of 0.0296 degrees in the range from 0 to 114 2$\ominus$ and a 2400 s acquisition time using CuKα radiation (λ = 1.54056 Å). The phases were identified by a Match! Software package (2003–2015 CRYSTAL IMPACT, Bonn, Germany) as follows: α-$GeO_2$ (SG No 136; PDF #35-0729); α—$GeO_2$ (SG No 136; PDF #21-0902); β-$GeO_2$ (SG No 154; PDF #43-1016); $Bi_2Ge_3O_9$ (SG No 215; PDF #43-0216); $Bi_2Ge_3O_9$ (SG No 176; PDF #43-0216); $Bi_4Ge_3O_{12}$ (SG No 220; PDF #34-0416); $Bi_2GeO_5$ (SG No 36; PDF #36-0289); and $Bi_{12}GeO_{20}$ (SG No 197; PDF #77-0556).

## 3. Results

### 3.1. Glass Samples

The synthesized samples had a ruby-red color, which became more saturated with an increase in the bismuth oxide content (Figure 1). Samples containing >40 mol% $Bi_2O_3$ had color inhomogeneity, probably caused by heterogeneous component distribution in the glass.

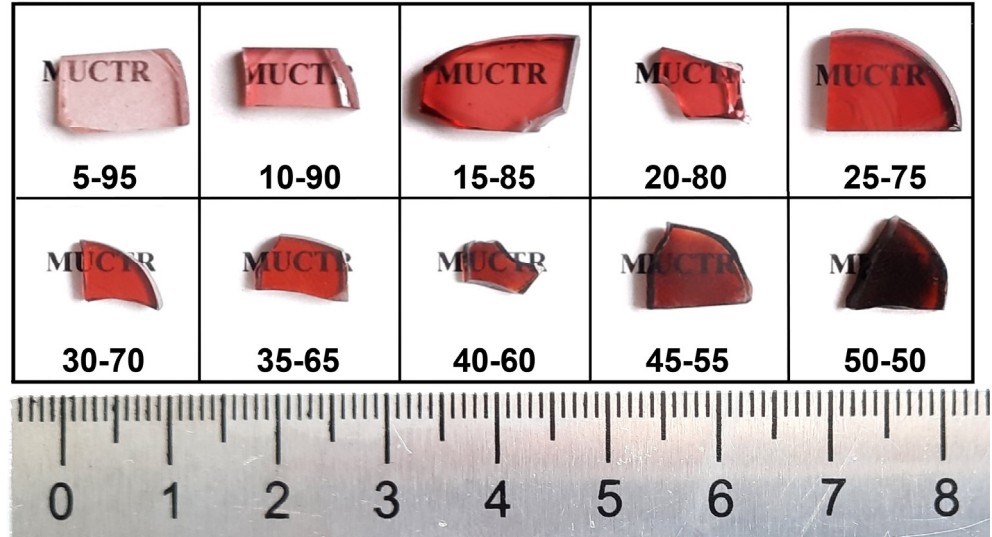

**Figure 1.** Photos of synthesized glasses. Here and after the numbers, refer to the sample ID.

The sample with the lowest content of bismuth oxide (5 mol%) had inclusions of bubbles due to the high melt viscosity at the synthesis temperature, and some of its properties were not studied. As a result, the 5-95 sample density was lower than the density of pure $GeO_2$. Glasses 50-50 were inclined to surface crystallization during melt casting, which contradicted the data of [2], in which $85.7Bi_2O_3$-$14.3GeO_2$ glasses were presented and similar synthesis conditions (temperature 1100–1200 °C, quenching on a metal substrate at room temperature) were reported for their production.

The results of the elemental analysis of the glasses showed that all samples contained an aluminum impurity (Table 1) due to the synthesis in corundum crucibles. A similar result was observed in [14]. With an increase in the bismuth oxide content, the amount of aluminum in the glass composition increased, which was explained by the chemical aggressiveness of the bismuth oxide melt towards the crucible material. Additionally, bismuth volatilized insignificantly during the synthesis, which was also described in the literature [15]. At the same time, the Bi/Ge ratio in our initial mixture and in the synthesized glass remained nearly unchanged.

**Table 1.** The composition of glasses according to the EDS data.

| Sample ID | Bi Content (mol%) | | Ge Content (mol%) | | O Content (mol%) | | Al Content (mol%) | |
|---|---|---|---|---|---|---|---|---|
| | Raw | EDS | Raw | EDS | Raw | EDS * | Raw | EDS |
| 10-90 | 6.25 | 8.51 ± 0.29 | 28.13 | 25.89 ± 0.13 | 65.62 | 65.18 | 0.00 | 0.42 ± 0.02 |
| 15-85 | 9.09 | 10.91 ± 0.37 | 25.76 | 22.35 ± 0.11 | 65.15 | 63.25 | 0.00 | 3.49 ± 0.14 |
| 20-80 | 11.76 | 10.52 ± 0.36 | 23.53 | 21.77 ± 0.11 | 64.71 | 65.97 | 0.00 | 1.74 ± 0.07 |
| 25-75 | 14.29 | 13.54 ± 0.46 | 21.43 | 18.21 ± 0.09 | 64.28 | 63.62 | 0.00 | 4.63 ± 0.18 |
| 30-70 | 16.67 | 15.43 ± 0.52 | 19.45 | 16.36 ± 0.08 | 63.88 | 63.30 | 0.00 | 4.90 ± 0.19 |
| 35-65 | 18.92 | 17.01 ± 0.58 | 17.57 | 14.22 ± 0.07 | 63.51 | 62.84 | 0.00 | 5.93 ± 0.23 |
| 40-60 | 21.05 | 18.61 ± 0.63 | 15.79 | 12.78 ± 0.07 | 63.15 | 62.58 | 0.00 | 6.03 ± 0.24 |
| 45-55 | 23.08 | 20.93 ± 0.71 | 14.10 | 11.35 ± 0.06 | 62.82 | 62.22 | 0.00 | 5.50 ± 0.21 |
| 50-50 | 25.00 | 22.80 ± 0.78 | 12.50 | 9.76 ± 0.05 | 62.50 | 61.90 | 0.00 | 5.54 ± 0.22 |

* Oxygen content was calculated as 100-$x$Bi-$y$Ge-$z$Al.

### 3.2. Glass Structure Characterization

Structural units in the glass network were characterized using Raman and IR spectroscopy (Figures 2 and 3). In the low-frequency region (<700 cm$^{-1}$) for the Raman spectra (Figure 2), the bands in the region of 500 cm$^{-1}$ characterized [GeO$_4$]-tetrahedra vibrations. Their intensity decreased with a reduction in germanium oxide concentration [2,16]. Additionally, in this region, there was a wide band at 600 cm$^{-1}$, related to vibrations of the Bi–O bond of [BiO$_6$]-octahedra [17]. It is interesting that for the 50-50 glass, the band at 395 cm$^{-1}$, associated with the bending of the O–Ge–O bridge bond [16], had the highest intensity in comparison with other glasses. It can be explained by the tendency of this glass to surface crystallize GeO$_2$ phases.

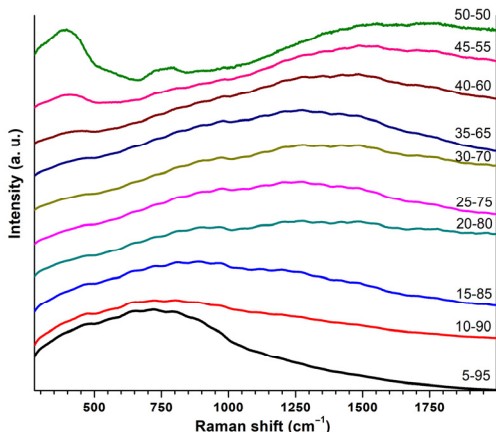

**Figure 2.** Raman spectra of synthesized glasses.

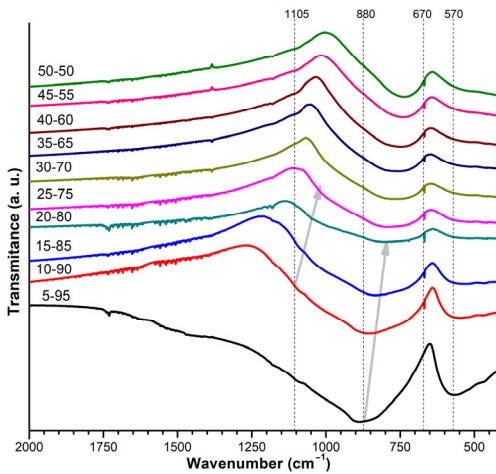

**Figure 3.** FT-IR spectra of synthesized glasses.

The bands in the high frequency region of the Raman spectra (>700 cm$^{-1}$) were assigned to [GeO$_4$]-tetrahedra vibrations with different numbers of non-bridging oxygen atoms, so-called Q$_n$-units, where *n* is the number of bridging oxygen atoms [18,19]. The growth of bismuth oxide content (Figure 2) resulted in the increasing intensity of the bands in the high-frequency region. This indicated an increase in the defectiveness of the glass structure.

The FT-IR spectra of the glasses (Figure 3) contained the main bands at 580, 670, 850, and 1105 cm$^{-1}$. The band at 580 cm$^{-1}$ referred to asymmetric stretching of the Ge–O–Ge bridge bond vibrations [19] and was observed for all glasses; its intensity decreased with increasing Bi$_2$O$_3$ content. The band at 670 cm$^{-1}$ was assigned to vibrations of Bi–O bonds in [BiO$_6$] structural units [20]. The band at 880 cm$^{-1}$ was assigned to Ge–O–Ge stretching [21].

The band at 1105 cm$^{-1}$ was assigned to vibrations of the Bi–O–Bi or Bi–O–Ge bond [20]. It should be noted that the bands at 880 and 1105 cm$^{-1}$ shifted to the low-frequency region with an increase in the bismuth oxide content, which indicated a weakening of the Ge–O bonds due to the incorporation of bismuth ions into the glass network. The FT-IR transmission spectra (Figure 3) confirmed the assumptions about the glass structure and were in agreement with the Raman spectra presented above.

### 3.3. DSC Characterization and Physical Properties

The glass transition temperatures ($T_g$) and maximum crystallization temperatures ($T_x$) of all samples (Table 2) were determined from DSC curves (Figures S1–S10).

**Table 2.** Glass characteristic temperatures *.

| Sample ID | $T_g$, °C | $T_{x1}$, °C | $T_{x2}$, °C | $T_{x3}$, °C |
|:---:|:---:|:---:|:---:|:---:|
| 5-95 | 470 | 631 | 662 | 690 |
| 10-90 | 460 | 651 | 719 | – |
| 15-85 | 461 | 650 | 707 | – |
| 20-80 | 469 | 696 | 744 | – |
| 25-75 | 470 | 663 | 689 | 721 |
| 30-70 | 473 | 633 | 712 | – |
| 35-65 | 478 | 647 | 663 | 692 |
| 40-60 | 469 | 624 | 657 | – |
| 45-55 | 441 | 518 | 575 | 654 |
| 50-50 | 450 | 548 | 598 | 657 |

*—the determination error for all characteristic temperatures was ±1 °C.

The presence of several crystallization temperatures was associated with the formation of various crystalline phases. The difference in the number of crystallization temperatures (2 or 3) for different compositions can be associated both with a change in the type of crystallizing phases and with a rather high heating rate of the samples during the DSC processing. The formation of the metastable $Bi_2GeO_5$ phase could be observed in the 600–650 °C temperature range, according to [22]. The crystallization temperature in the region of 650–700 °C may correspond to the transition of the metastable $Bi_2GeO_5$ phase to the stable $Bi_4Ge_3O_{12}$ with the eulytite structure [23]. The shift of the crystallization temperatures of the same phase towards high values for glasses with a $Bi_2O_3$ content <30 mol% is explained by the lower tendency of these glasses to crystallize (Figure 4).

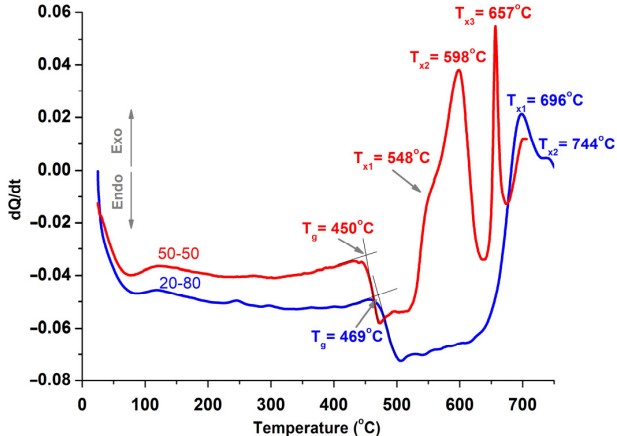

**Figure 4.** DSC curves of 20-80 and 50-50 synthesized glasses.

The density and refractive index of glasses (Table 3) expectedly increased with the growth in bismuth oxide content. The obtained results correlated with the data [5,6]. A slight decrease in the density and refractive index can be explained by the entry of aluminum oxide from the crucibles into the glasses.

**Table 3.** Density and refractive index values of glasses.

| Sample | Density (g/cm$^3$) | Refractive Index at 589 nm |
|---|---|---|
| 0-100 * | 4.25 | |
| 5-95 | 4.085 ± 0.005 | 1.68 ± 0.01 |
| 10-90 | 4.645 ± 0.005 | 1.76 ± 0.01 |
| 15-85 | 5.185 ± 0.005 | 1.80 ± 0.04 |
| 20-80 | 5.370 ± 0.005 | 1.86 ± 0.06 |
| 25-75 | 5.945 ± 0.005 | 2.06 ± 0.06 |
| 30-70 | 6.015 ± 0.005 | 2.08 ± 0.02 |
| 35-65 | 6.330 ± 0.005 | 2.10 ± 0.04 |
| 40-60 | 6.685 ± 0.005 | 2.12 ± 0.02 |
| 45-55 | 6.760 ± 0.005 | 2.14 ± 0.04 |
| 50-50 | 7.085 ± 0.005 | 2.14 ± 0.04 |
| 100-0 * | 8.90 | |

\* Values are presented for pure oxides.

### 3.4. Spectral-Luminescent Properties

The absorption spectra of glasses (Figure 5) exhibited a characteristic shoulder at 500 nm associated with BACs [7–9]. The absorption coefficient in this region increased with the growth of the bismuth oxide content.

Similarly, with an increase in the bismuth oxide content, the short-wavelength absorption edge shifted from 340 nm (Sample ID 5-95) to 425 nm (Sample ID 50-50). This shift was due to the fact that the optical band gap of bismuth (III) oxide is smaller than that of germanium oxide (5.63 eV) and ranges from 2.5 to 3.2 eV for various $Bi_2O_3$ polymorphs [14].

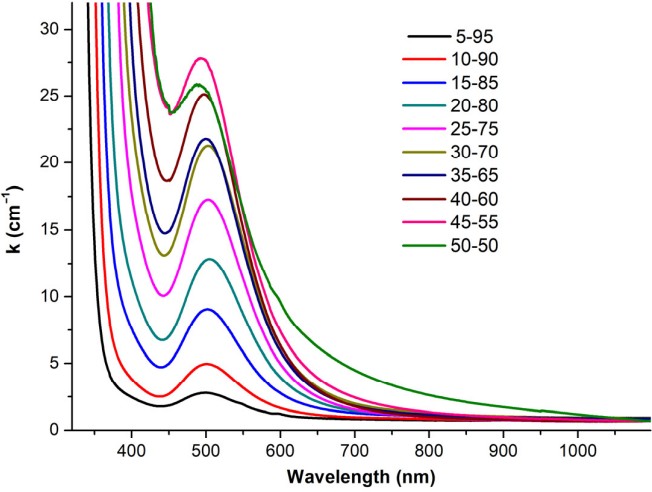

**Figure 5.** Optical absorption spectra of synthesized glasses.

To determine the width of the optical energy gap ($E_g$) of glasses, the Tauc method was used (Figure 6, Table 4).

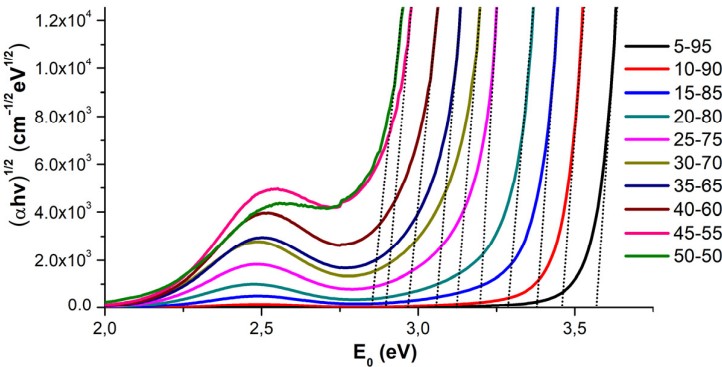

**Figure 6.** Tauc's plots.

**Table 4.** Energy band gap of bismuth-germanate glasses.

| Sample ID | $E_g$ (eV) * |
|:---:|:---:|
| 5-95 | 3.54 |
| 10-90 | 3.47 |
| 15-85 | 3.35 |
| 20-80 | 3.28 |
| 25-75 | 3.20 |
| 30-70 | 3.08 |
| 35-65 | 3.04 |
| 40-60 | 2.95 |
| 45-55 | 2.86 |
| 50-50 | 2.82 |

*—the determination error for $E_g$ was $\pm 0.02$ eV.

Under excitation of photoluminescence (PL) at wavelengths of 405, 425, 525, 650, and 805 nm for 5-95 samples (Figure 7), it was found that 450 nm was the optimal excitation for BACs (see Figure S19). We observed that a green laser (525 nm) action led to strong heating of the glasses, which significantly decreased the PL intensity.

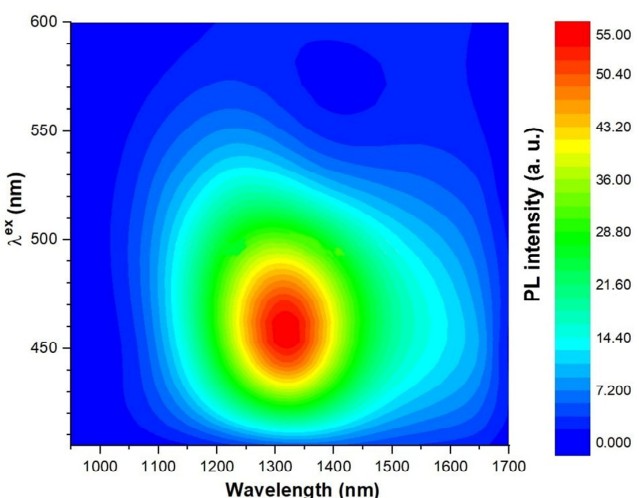

**Figure 7.** Luminescence spectrum of 5-95 glass at $\lambda^{ex}$ = 400–600 nm.

The PL spectra of glasses at $\lambda^{ex}$ = 450 nm (Figure 8) represented a wide band in the near IR region. As can be seen, the luminescence region corresponded to the data of [7–10], which additionally confirms the presence of BACs in glasses. For tested glasses, when the bismuth oxide content increased, the PL intensity became lower due to concentration quenching.

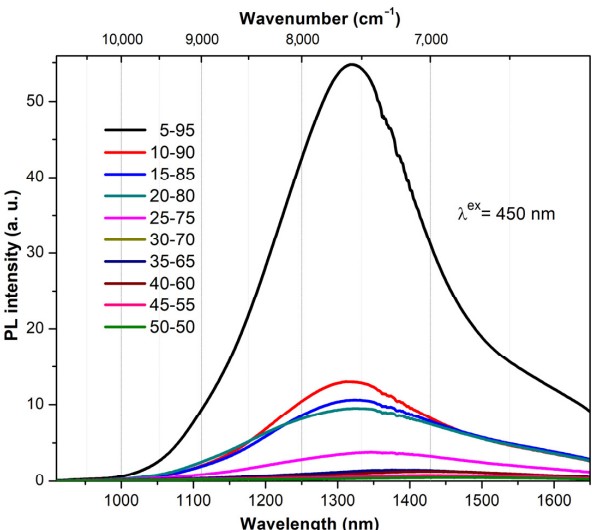

**Figure 8.** Photoluminescence spectra of glasses ($\lambda^{ex}$ = 450 nm).

Sample 5-95 demonstrated the highest PL intensity (Figure 9). The observed broadband luminescence was attributed to low-valence forms of bismuth ($Bi^{n<2+}$) in the BACs [11,24,25].

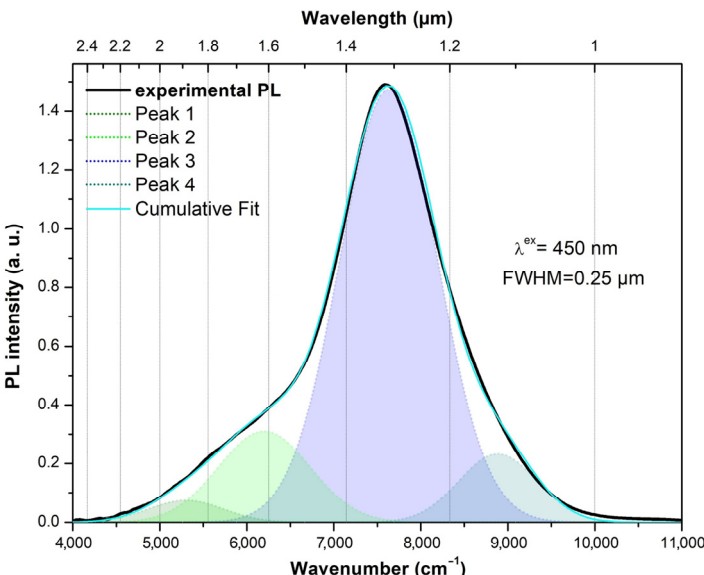

**Figure 9.** Photoluminescence spectrum of 5-95 glass ($\lambda^{ex}$ = 450 nm).

## 4. Discussion

Analysis of the optical absorption and luminescence spectra of the synthesized glasses ($\lambda^{ex}$ = 450 nm) showed the presence of BACs, the number of which increased with the bismuth oxide total concentration growth. The contour of the PL spectrum in the IR region was represented by a superposition of several bands, whose maxima, determined from the Gaussian components, were located at wavelengths ~1125, 1310, 1615, and 1885 nm (Figure 9). According to [26,27], the bands at 1125 and 1310 nm corresponded to the $^3P_1 \rightarrow {}^3P_0$ transitions for the $Bi^+$ ion and the $^2D_{3/2} \rightarrow {}^4S_{3/2}$ transitions for $Bi^0$, respectively.

At the same time, it was shown in [11] that BACs were not individual low-valence bismuth ions, but a complex system of cations and an oxygen vacancy. In this case, both bands at 1125 nm (Peak 4 in Figure 9) and 1310 nm (Peak 3 in Figure 9) belonged to oxygen-deficient centers $=Bi\cdots Ge\equiv$. Thus, the difference in the band position was caused by the presence or absence of aluminum ions in the second coordination sphere of the BACs, respectively [11]. Previously, for the samples with a high content of $Bi_2O_3$ (>20 mol%), the luminescence was observed in the longer wavelength part of the spectrum (1800–3000 nm). It was supposed that this luminescence could be attributed to the formation of $Bi_5^{3+}$ cluster centers [28] or oxygen-deficient centers $=Bi\cdots Bi=$ [11]. We assume that in our glasses two types of luminescent BACs were formed: namely, $=Bi\cdots Ge\equiv$ (~1125 and ~1310 nm) (Peaks 4 and 3 in Figure 9) and $=Bi\cdots Bi=$ (1615 and 1885 nm) (Peaks 2 and 1 in Figure 9) in a smaller amount. $Bi_2O_3$ content growth led to an increase in the amount of $=Bi\cdots Bi=$ type centers and to PL decreasing in the ~1300 nm region. This BACs transformation was in good agreement with the structural analysis data. The shift of the vibration bands towards low frequencies at the bismuth oxide content growth indicated an increase in the Ge–O and Bi–O bond lengths, which in turn resulted in the formation of $=Bi\cdots Bi=$ centers having shorter bond lengths than the $=Bi\cdots Ge\equiv$ centers [11].

The glass transition temperatures of bismuth-containing glasses were lower compared to the temperature of pure $GeO_2$ glass (519 °C [29]), probably due to a decrease in melt viscosity upon the introduction of $Bi_2O_3$. The resulting range of $T_g$ values (440–480 °C) was in good agreement with the data previously reported [30–33].

The DSC data showed the possibility of crystallization of several phases in glasses, and the set of crystalline phases varied for different glass compositions. The heat treatment of samples at 600 °C showed (Figure 10 and Figures S11–S17) that predominantly $\alpha$-$GeO_2$

and β-GeO$_2$ phases crystallized, accompanied by a certain amount of the Bi$_4$Ge$_3$O$_{12}$ phase in samples containing up to 20 mol% Bi$_2$O$_3$. The crystallization peaks of all phases for these compositions were weakly separated (Figures S1–S3), which indicated the almost simultaneous beginning of their crystallization process in glass. The simultaneous existence of both modifications of crystalline GeO$_2$ correlated well with the metastable phase diagram [3], in which ~600 °C served as the transition temperature between α-GeO$_2$ and β-GeO$_2$ polymorphs. The Bi$_2$GeO$_5$ phase, noted in the same phase diagram, was unstable in this concentration range, as shown in [33], and appeared only in trace amounts in the 5-95 sample according to XRD patterns (see Figure 10).

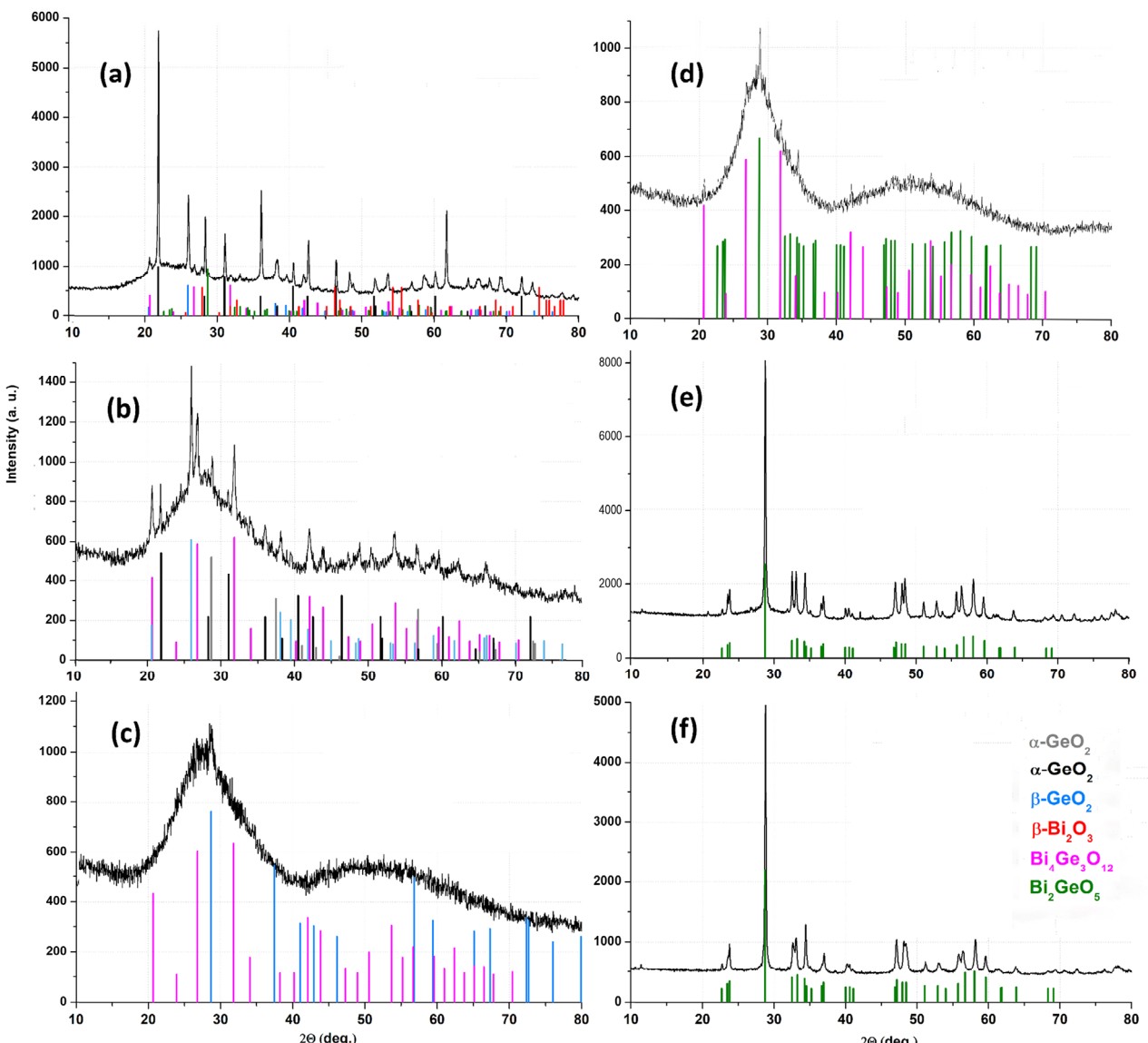

**Figure 10.** XRD patterns of crystallized glasses (**a**) 5-95, (**b**) 10-90, (**c**) 15-85, (**d**) 40-60, (**e**) 45-55, and (**f**) 50-50 heat-treated at 600 °C for 2 h (for details, see Figures S11–S13,S15–S17).

Crystallization in the 550–640 °C temperature range leads to the formation of the Bi$_4$Ge$_3$O$_{12}$ phase [30–32], alone or together with other phases for glass compositions containing 10–40 mol% Bi$_2$O$_3$. Therefore, the crystallization peaks belonging to the 650–663 °C range can be associated with the maximum crystallization temperature of the Bi$_4$Ge$_3$O$_{12}$ phase (Figure 11). The crystallization peaks in the 690–744 °C range can be associated with the crystallization temperature of the β-GeO$_2$ phase for samples with a high content of GeO$_2$ or other phases for samples with a high content of Bi$_2$O$_3$.

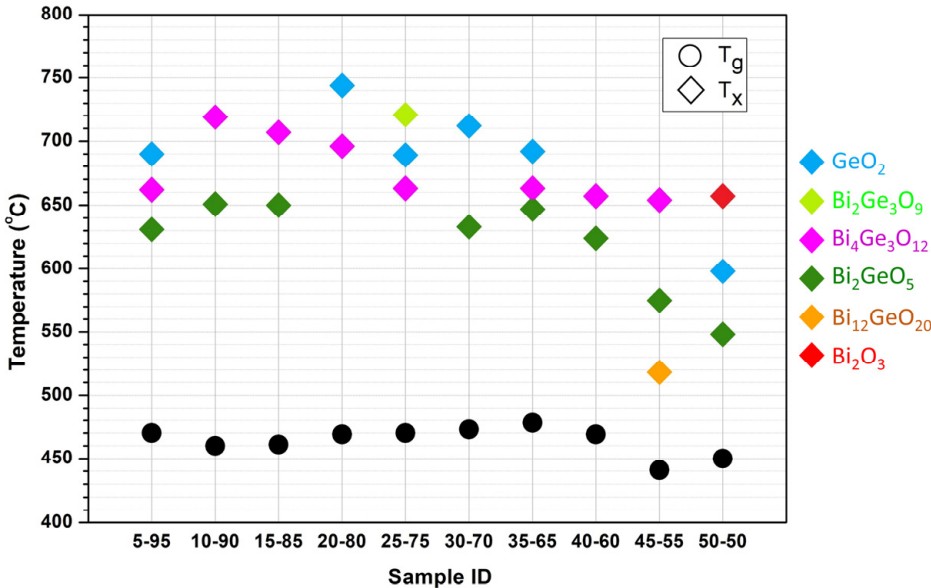

**Figure 11.** Glass transition and maximum crystallization temperatures of different phases formed from glasses with varied $Bi_2O_3$ composition.

It is known that the set of crystalline phases in crystallized glasses changes as the content of $Bi_2O_3$ increases ($\geq$20 mol%). $Bi_4Ge_3O_{12}$ and $Bi_2GeO_5$ become the main phases [34,35]. Therefore, despite the XRD amorphous halo for our glasses containing 20–35 mol% $Bi_2O_3$ and heat-treated at 600 °C (Figure S14), it can be assumed that the crystallization proceeded similarly in our samples. Consequently, for the 624–647 °C range, the crystallization peaks belonged to the $Bi_2GeO_5$ phase. This assumption was also supported by the fact that the $Bi_2GeO_5$ phase disappearance was noted in [36] when the heat treatment temperature increased above 640 °C. Heat-treatment of 25-75 glass (Figure S18) at 690 °C led to the $Bi_4Ge_3O_{12}$ and $Bi_2Ge_3O_9$ phases' formation in agreement with [33,34]. An increase of the heat treatment temperature to 720 °C for the 25-75 glass led to an insignificant decrease in the amount of the $Bi_2Ge_3O_9$ phase, whose composition corresponded to that of glass; therefore, the crystallization maximum at 690 °C on the DSC curve corresponded to the $Bi_2Ge_3O_9$ phase formation. The formation of phases' mixtures during 25-75 glass crystallization corresponded to the cross sections of the Bi–Ge–O phase diagram [37], where in the region of 25 mol% $Bi_2O_3$ we observed $S_{Bi_4Ge_3O_{12}} - S_{Bi_2Ge_3O_9} - V$ bivariant equilibrium. The same phase equilibrium explains the absence of the β-$GeO_2$ phase in the crystallized glasses, which demonstrated a weak crystallization peak at 721 °C on the DSC curve.

The crystallization of 45-55 and 50-50 glasses should be discussed in detail. These glasses were inclined to crystallization in the glass casting process already. Therefore, there was a possibility of the spontaneous nuclei of crystalline phases' existence in glasses that were not determined by XRD. The crystallization maximum at temperatures of 575–598 °C belonged probably to the $Bi_2GeO_5$ phase since heat treatment at 600 °C led to the formation of this particular phase as the main one in 45-55 glass and the only one in 50-50 glass. To confirm these conclusions and to identify the crystallization peaks in the 518–548 °C range, which were not observed for the rest of the glass compositions, additional annealing of 45-55 and 50-50 glasses was carried out.

For the 45-55 sample (Figure 12), annealing at 520 °C led to the formation of the $Bi_{12}GeO_{20}$ phase together with the $Bi_2GeO_5$ and $Bi_4Ge_3O_{12}$ phases. The composition of the $Bi_{12}GeO_{20}$ phase corresponds to the molar composition of 85.7$Bi_2O_3$-14.3$GeO_2$, which is quite far from the original 45-55 glass composition. However, if we consider the Bi–Ge–O phase diagram cross sections [37] at temperatures close to the heat-treatment temperature (517 °C, 596 °C), it becomes clear that the 45-55 composition is in the range of monovariant equilibrium $S_{Bi_{12}GeO_{20}} - S_{Bi_4Ge_3O_{12}} - S_{Bi_2GeO_5} - V$. Annealing of 45-55 glass at 580 °C led to

the crystallization of only two phases: $Bi_2GeO_5$ (basic) and $Bi_4Ge_3O_{12}$ (Figure 12). The area of $S_{Bi_{12}GeO_{20}} - S_{Bi_4Ge_3O_{12}} - S_{Bi_2GeO_5} - V$ monovariant equilibrium narrowed at temperature increases from 799 K to 850 K in the Bi–Ge–O diagram cross section [37]. As a result, the 45-55 glass composition moved to the region of $S_{Bi_4Ge_3O_{12}} - S_{Bi_2GeO_5} - V$ bivariant equilibrium. Thus, for 45-55 glass, the exothermic peak at 518 °C referred to the maximum crystallization temperature of the $Bi_{12}GeO_{20}$ phase, while the peak at 575 °C referred to the $Bi_2GeO_5$ phase.

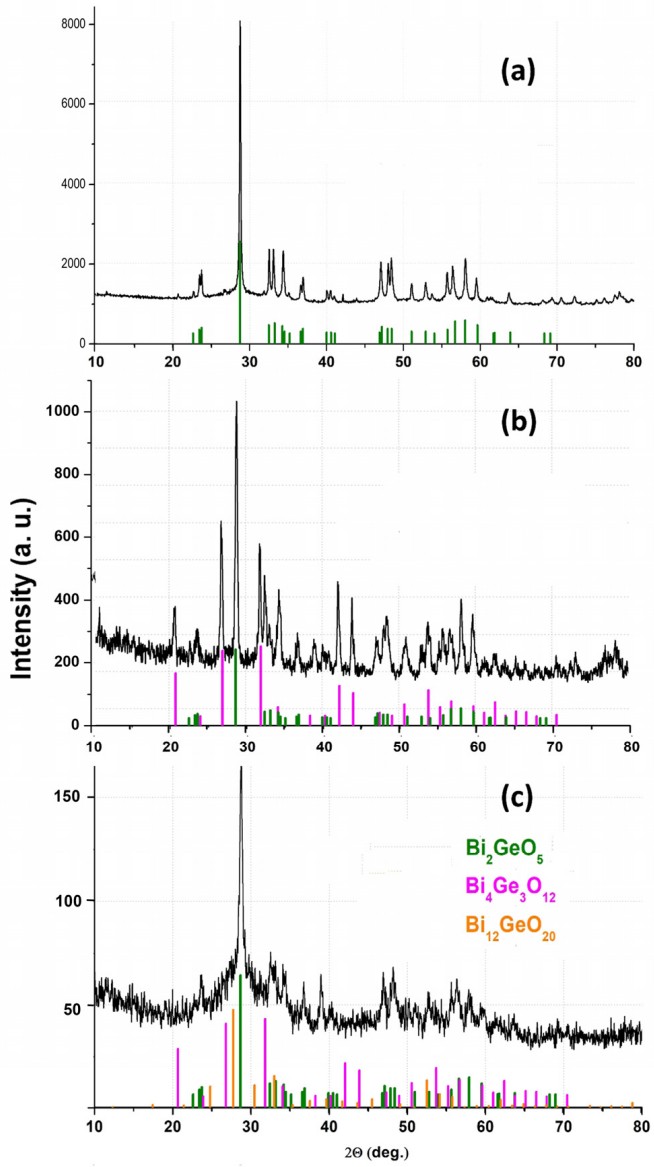

**Figure 12.** XRD patterns of crystallized 45-55 glasses heat-treated for 2 h at different temperatures: (**a**) 600 °C, (**b**) 580 °C, and (**c**) 520 °C.

Heat treatment of 50-50 glass at 550 °C led to the single $Bi_2GeO_5$ phase formation corresponding to the glass composition (Figure 13). The increase in heat-treatment temperature to 600 °C led to the appearance of the mixture of $Bi_2GeO_5$ and β-$GeO_2$ phases. The further temperature rise to 750 °C caused the formation of the mixture of $Bi_2GeO_5$, β-$GeO_2$, and β-$Bi_2O_3$ phases. Thus, the maximum crystallization temperatures of 548, 598, and 657 °C corresponded to the formation of $Bi_2GeO_5$, β-$GeO_2$, and β-$Bi_2O_3$ phases, respectively. The formation of the β-$GeO_2$ crystalline phase at high bismuth concentrations in the 50-50 sample could be caused by composition fluctuations in the initial glass.

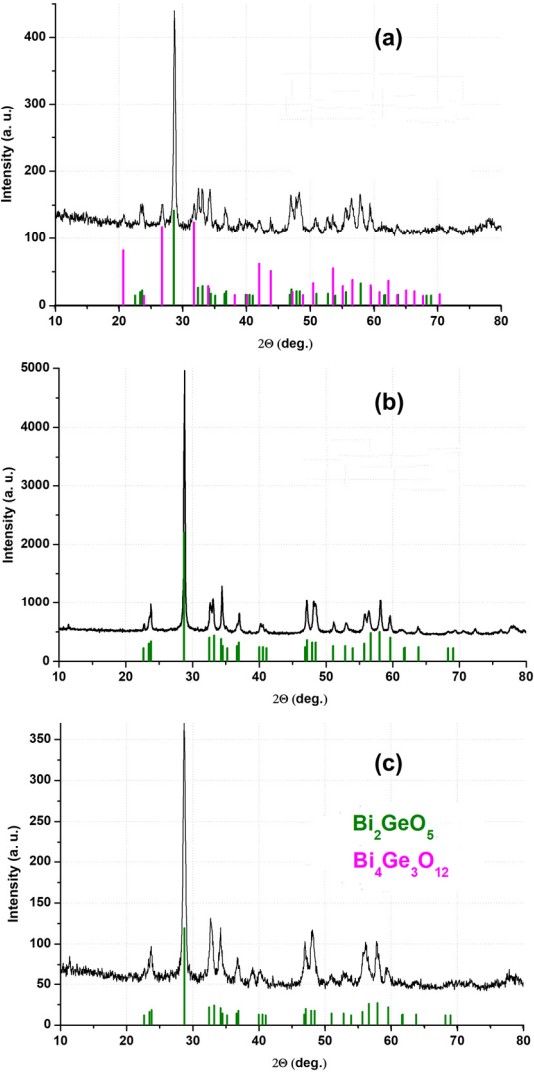

**Figure 13.** XRD patterns of crystallized 50-50 glasses heat-treated for 2 h at different temperatures: (**a**) 750 °C, (**b**) 600 °C, and (**c**) 550 °C.

Summarizing the crystallization data, we can say that the crystallization temperatures of bismuth-germanate phases correlate well with the amount of bismuth in their composition. The decrease in $Bi_2O_3$ content in the row of individual compounds $Bi_{12}GeO_{20} \rightarrow Bi_2GeO_5 \rightarrow Bi_4Ge_3O_{12} \rightarrow Bi_2Ge_3O_9$ (85.7–50–40–25 mol%) results in an increase in the maximum crystallization temperatures of the corresponding phases.

## 5. Conclusions

To fill the gaps in fundamental data for the first time, we investigated bismuth and germanium oxide-based glasses in a wide concentration range, with special emphasis on high $Bi_2O_3$ concentrations up to 50 mol%. We succeeded in decreasing the synthesis temperature from 1300 to 1100 °C. Glasses based on bismuth oxide and germanium oxide demonstrated a strong dependence of their structure and properties on the $Bi_2O_3/GeO_2$ ratio. An increase in the bismuth oxide concentration led to an increase in the number of non-bridging oxygen ions and a weakening of the Ge–O bonds. Such a rearrangement of the glass structure contributed to the destruction of $=Bi\cdots Ge\equiv$ bismuth luminescent centers and the formation of $=Bi\cdots Bi=$ luminescent centers, which led to a weakening of the PL intensity in the region of ~1300 nm. The results of glass crystallization depended on the $Bi_2O_3$ oxide content: the higher the $Bi_2O_3$ concentration in a crystalline phase, the lower the temperature of its formation.

**Supplementary Materials:** The following supporting information can be downloaded at: https://www.mdpi.com/article/10.3390/ceramics6030097/s1, Figure S1: DSC curve of synthesized 5-95 glass; Figure S2: DSC curve of 10-90 glass; Figure S3: DSC curve of synthesized 15-85 glass; Figure S4: DSC curve of synthesized 20-80 glass; Figure S5: DSC curve of synthesized 25-75 glass; Figure S6: DSC curve of synthesized 30-70 glass; Figure S7: DSC curve of synthesized 35-65 glass; Figure S8: DSC curve of synthesized 40-60 glass; Figure S9: DSC curve of synthesized 45-55 glass; Figure S10: DSC curve of synthesized 50-50 glass; Figure S11: XRD patterns of 5-95 glass heat-treated at 600 °C; Figure S12: XRD patterns of 10-90 glass heat-treated at 600 °C; Figure S13: XRD patterns of 15-85 glass heat-treated at 600 °C; Figure S14: XRD patterns of 20-80, 25-75, 30-70, and 35-65 glasses heat-treated at 600 °C; Figure S15: XRD patterns of 40-60 glass heat-treated at 600 °C; Figure S16: XRD patterns of 45-55 glass heat-treated at 600 °C; Figure S17: XRD patterns of 50-50 glass heat-treated at 600 °C; Figure S18: XRD patterns of 25-75 glass heat-treated at 600, 690, and 720 °C; Figure S19: Excitation and emission spectra of 5-95 glass ($\lambda^{ex}$ = 450 nm).

**Author Contributions:** Conceptualization, I.S. and I.A.; methodology, K.S.; software, K.N.; validation, K.S., I.S. and I.A.; formal analysis, M.U.; investigation, K.S., I.S., A.P., M.U., K.B. and R.A.; resources, K.N., K.B. and I.A.; data curation, K.S. and I.S.; writing—original draft preparation, K.S. and I.S.; writing—review and editing, I.S., R.A. and I.A.; visualization, K.S.; supervision, I.A.; project administration, I.S.; funding acquisition, R.A. All authors have read and agreed to the published version of the manuscript.

**Funding:** The research was financially supported by the Ministry of Science and Higher Education of Russia through the project FSSM-2020-0005.

**Institutional Review Board Statement:** Not applicable.

**Informed Consent Statement:** Not applicable.

**Data Availability Statement:** Not applicable.

**Conflicts of Interest:** The authors declare no conflict of interest.

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
