# Peer review of "Bismuth-Germanate Glasses: Synthesis, Structure, Luminescence, and Crystallization"

_ceramics, doi:10.3390/ceramics6030097_

Round 1

Reviewer 1 Report (Previous Reviewer 2)

The manuscript is of interest for the journal and suits it very well. The work presents a concise study on the glass system Bi2O3-GeO2, demontstrating optical properties of glasses in a wide range of ratios (5 to 50 mol. % of Bi2O3). Moreover, the authors demonstrate preparation of such glasses at lower temperature, at 1100 C, instead of 1300 C.

In comparison with the first version, the present version of this manuscript is indeed improved and reads better and more solid. It can be accepted after additional revision, mainly of technical character. Please see comments and suggestions that should help the authors improve it:

1\ Fig.1:  Scale bar is recommended (or a picture of an object like a coin or ruler), so that the readers can see the size of produced crystals.

2\ Fig.10: The authors have to add labels (a)-(f) and describe them in the caption.

3\ Fig.12: The authors have to add labels (a)-(c) and describe them in the caption.

3\ Fig.13: The authors have to add labels (a)-(c) and describe them in the caption.

1\ Line 322: probably, instead of beaks ,the authors may wish to use "gaps"

2\ Line 332: the newly added sentence is not well connected to the previous sentence (last sentence of the Conclusion).

3\ Lines 208-220: the newly added paragraph needs minor check and polishing.

Author Response

Dear Reviewer?

Thanks a lot for the useful comments. We made all the correction.

1\ Fig.1:  Scale bar is recommended (or a picture of an object like a coin or ruler), so that the readers can see the size of produced crystals.

Done

2\ Fig.10: The authors have to add labels (a)-(f) and describe them in the caption.

Done

3\ Fig.12: The authors have to add labels (a)-(c) and describe them in the caption.

Done

3\ Fig.13: The authors have to add labels (a)-(c) and describe them in the caption.

Done

Comments on the Quality of English Language

1\ Line 322: probably, instead of beaks ,the authors may wish to use "gaps"

Agreed. Corrected.

2\ Line 332: the newly added sentence is not well connected to the previous sentence (last sentence of the Conclusion).

Corrected. Move the sentence to the bottom of the Conclusions because it is the very important result of the research.

3\ Lines 208-220: the newly added paragraph needs minor check and polishing.

Agreed. Corrected. Tried to improve.

Reviewer 2 Report (New Reviewer)

Dear Editor,

Dear authors,

This manuscript is nicely written and might be interesting for the readership of Journal Ceramics. However, it can be accepted for publication after minor revision. Here are my comments.

Comment 1: English language corrections are required.

Comment 2: Please indicate which chemicals you used during the synthesis (Line 53).

Comment 3: Please indicate the conditions under which you recorded the samples for XRD analysis (Line 87). After that, list all the card numbers with the space groups in which the identified phases crystallize. For example: Bi2GeO5 (SG No xx; PDF #36-028922)

Comment 4: Please indicate in Table 3 (Line 165) (first and last row) what are the theoretical densities for Bi2O3 and GeO2, so that we can see the changes in density.

Comment 5: The intensity of the XRD analysis is of great importance for this and many other XRD analyses. Based on the height and width of the peaks from the pictures, one can see their degree of crystallization (of course, who knows how to look and calculate). Please do not make such elementary mistakes. Please put Intensity (a.u) on the ordinate (y-axis) in each image related to XRD analysis, remove the card numbers you used to identify the phases and leave only the name of the phase.

Minor editing of English language required.

Author Response

Dear Reviewer

Thanks a lot for the fruitful comments.

We made all corrections.

Comment 1: English language corrections are required.

We made our best to improve English

Comment 2: Please indicate which chemicals you used during the synthesis (Line 53).

Added the information

Comment 3: Please indicate the conditions under which you recorded the samples for XRD analysis (Line 87). After that, list all the card numbers with the space groups in which the identified phases crystallize. For example: Bi2GeO5 (SG No xx; PDF #36-028922)

We added the details of X-ray measurements

The structure of the crystalline phases was determined by X-ray diffraction using an Equinox-2000 diffractometer (Inel SAS, France) with linear CCD detector with the step 0.0296 degree in the range from 0 to 114 2Q and 2400 sec acquisition time using CuKα radiation (λ=1.54056 Å). The phases were identified by a Match! Software package (2003-2015 CRYSTAL IM-PAC T, Bonn, Germany) as follows: -GeO2 (SG No 136; PDF #35-0729); --GeO2 (SG No 136; PDF #21-0902); -GeO2 (SG No 154; PDF #43-1016); Bi2Ge3O9 (SG No 215; PDF #43-0216); Bi2Ge3O9 (SG No 176; PDF #43-0216); Bi4Ge3O12 (SG No 220; PDF #34-0416); Bi2GeO5 (SG No 36; PDF #36-0289); Bi12GeO20 (SG No 197; PDF #77-0556).

Comment 4: Please indicate in Table 3 (Line 165) (first and last row) what are the theoretical densities for Bi2O3 and GeO2, so that we can see the changes in density.

Added

Comment 5: The intensity of the XRD analysis is of great importance for this and many other XRD analyses. Based on the height and width of the peaks from the pictures, one can see their degree of crystallization (of course, who knows how to look and calculate). Please do not make such elementary mistakes. Please put Intensity (a.u) on the ordinate (y-axis) in each image related to XRD analysis, remove the card numbers you used to identify the phases and leave only the name of the phase.

Added the Intensity scale in a.u.

This manuscript is a resubmission of an earlier submission. The following is a list of the peer review reports and author responses from that submission.

Round 1

Reviewer 1 Report

Reviewers comments:

This manuscript describes simple synthesis of bismuth-germanate glasses through melt-quenching technique and their characterization. Authors have varied the ratio of Bi2O3 and GeO2 from 5-50 to study the effect of crystallization within glass as a function of temperature.  It is nicely written and might be interesting for the readership of Journal Ceramics. However, it can be accepted for publication after major revision. Here are my comments.

Comment 1: English language corrections are required.

Comment 2: Figure quality should be improved.

Comment 3: What is novelty statement of this work. Novelty statement is missing.

Comment 4: Check the manuscript thoroughly for grammatical and spelling errors. Please correct the text “characterisation” (section 3.2)

Comment 5: Authors are encouraged to measure the TEM and HRTEM analysis and elemental mapping to determine the particle size and crystalline phases.

Comment 6: Authors are encouraged to include some interesting applications (e.g. cut off filters, nonlinear optics) of as synthesized glasses. The effect of size quantization on the catalytic activity can be interesting particularly for photocatalysis or biological applications such as antibacterial etc.              

English language corrections required. 

Author Response

Comment 1: English language corrections are required.

Reply

We made our best to improve the language.

Comment 2: Figure quality should be improved.

Reply

Corrected

Comment 3: What is novelty statement of this work. Novelty statement is missing.

Reply

Comparing with the previous papers dealing with bismuth and germanium oxide-based glasses we enlarge the range of Bi2O3 concentration up to 50 mol% and decrease the synthesis temperature from 1300 to 1100 °C. We added these distinguishing features to the Abstract and in the Conclusions section.

Comment 4: Check the manuscript thoroughly for grammatical and spelling errors. Please correct the text “characterisation” (section 3.2)

Reply

Done. Corrected.

Comment 5: Authors are encouraged to measure the TEM and HRTEM analysis and elemental mapping to determine the particle size and crystalline phases.

Reply

We focused our attention on the problem of glass crystallization and phase formation. X-ray measurements was enough to answer the questions and to determine the dependence between glass composition, temperature and phase compositions (including XFS/EDS data).

Comment 6: Authors are encouraged to include some interesting applications (e.g. cut off filters, nonlinear optics) of as synthesized glasses. The effect of size quantization on the catalytic activity can be interesting particularly for photocatalysis or biological applications such as antibacterial etc.    

Reply

We added the examples of possible application of our glasses in the Discussion section.

Reviewer 2 Report

The manuscript fits the journal very well. It analyzes the Bi2O3-GeO2 system, covering a wide range of compositions.

The manuscript, however, cannot be accepted in its present form and should be revised, after which it can be reconsidered. Below are comments and suggestions that should help the authors improve it:

1\ lines 7,11: the same email address is mentioned (which is meaningless)

2\ line 18: probably should be :  "resulted in an increase of density and refractive index" ?

3\ line 20: instead of "resulted to" should be "resulted in"

4\ line 30: "but it can vary from" (what is meant by "it"? who or what can vary from ...?)

5\ The authors should clarify what original and novel is presented by the manuscript. 1-2 sentences on this matter should be added in the end of Introduction.

6\ lines 52, 54: instead of "minutes" and "hours", it's better to use "min" and "h"

7\ line 83: decimal point should be used instead of comma in 1.54 A

8\ line 174: Tauc's plots? (add "s")

9\ Fig.10: the figure has too small features. Probably it should be split into 2-3 figures, which can be then enlarged

10\ Figures 12,13: what is shown along the Y-axis? arbitrary units?

See comments above

Author Response

The manuscript, however, cannot be accepted in its present form and should be revised, after which it can be reconsidered. Below are comments and suggestions that should help the authors improve it:

1\ lines 7,11: the same email address is mentioned (which is meaningless)

Reply

The same email address is mentioned to clarify the email address for correspondence.

2\ line 18: probably should be :  "resulted in an increase of density and refractive index" ?

Reply

Agreed. Corrected.

3\ line 20: instead of "resulted to" should be "resulted in"

Reply

Agreed. Corrected.

4\ line 30: "but it can vary from" (what is meant by "it"? who or what can vary from ...?)

Reply

Agreed. Corrected.

5\ The authors should clarify what original and novel is presented by the manuscript. 1-2 sentences on this matter should be added in the end of Introduction.

Reply

Comparing with the previous papers dealing with bismuth and germanium oxide-based glasses we enlarge the range of Bi2O3 concentration up to 50 mol% and decrease the synthesis temperature from 1300 to 1100 °C. We added these distinguishing features to the Abstract and Conclusions.

6\ lines 52, 54: instead of "minutes" and "hours", it's better to use "min" and "h"

Reply

Agreed. Corrected.

7\ line 83: decimal point should be used instead of comma in 1.54 A

Reply

Agreed. Corrected.

8\ line 174: Tauc's plots? (add "s")

Reply

Agreed. Corrected.

9\ Fig.10: the figure has too small features. Probably it should be split into 2-3 figures, which can be then enlarged.

Reply

For each sample the XRD pattern is presented in the large scale in the Supplementary section.

10\ Figures 12,13: what is shown along the Y-axis? arbitrary units?

Reply

The X-ray intensity has no meaning for this analysis. So, we deleted the units of Y-scale.

Reviewer 3 Report

The authors report the optical and luminescence behavior of Bi2O3-GeO2 glasses.

 Compared with the previous papers, such as Laser Phys. 23 (2013) 105812 (Ref. 34) and Opt. Mater. 34 (2012) 675, no remarkable novelty and improvement is observed.

The paper can not be accepted in present form. Not only abstract but also the main text and figures should be revised so that the differences are clear.

 Additional comments>

Since units of nm, cm-1, and eV are mixed, unify the units in the figure to eV (Figures 5, 7, 8, 9).

Also, the K and °C units are mixed in the text. Revise them.

In Table 1, data of 5-95 are missing.

Significant figures of density in Table 3 are wrong. It is inherently wrong to display a value smaller than the error bars.

Density of pure GeO2 glass should be added in Table 3 in order to check the validity of chemical composition, although EDS measurement were done.

Excitation spectra of Fig. 8 should be shown.

The PL peak deconvolution (Fig.9) is just a presentation. Add discussion about the relationship between the side peak and chemical compositions.

What does crystalline phase mean in Figure 11? The heat treatment gives the impression of a precipitation of single phase, even though no single phase has appeared in several glass-ceramics. It is confusing and should be removed.

Author Response

Compared with the previous papers, such as Laser Phys. 23 (2013) 105812 (Ref. 34) and Opt. Mater. 34 (2012) 675, no remarkable novelty and improvement is observed.

Reply

Comparing with the previous papers dealing with bismuth and germanium oxide-based glasses we enlarge the range of Bi2O3 concentration up to 50 mol% and decrease the synthesis temperature from 1300 to 1100 °C.

Comment #0

The paper can not be accepted in present form. Not only abstract but also the main text and figures should be revised so that the differences are clear.

Reply

We changed the main text, Abstract and Conclusions

 Additional comments>

Comment #1

Since units of nm, cm-1, and eV are mixed, unify the units in the figure to eV (Figures 5, 7, 8, 9).

Reply

Agreed. Corrected.

Comment #2

Also, the K and °C units are mixed in the text. Revise them.

Reply

Agreed. Corrected.

Comment #3

In Table 1, data of 5-95 are missing.

Reply

The 5-95 sample had a lot of bubbles inside. So, when a plane-parallel sample was obtained by mechanical polishing, the bubbles inside the glass turned out to the surface. The presence of pores on the surface is an additional source of undesirable impurities, for this reason, element analysis of the 5-95 glass sample wasn’t carried out.

Comment #4

Significant figures of density in Table 3 are wrong. It is inherently wrong to display a value smaller than the error bars.

Reply

Sorry for the misprint. Corrected the accuracy value.

Comment #5

Density of pure GeO2 glass should be added in Table 3 in order to check the validity of chemical composition, although EDS measurement were done.

Reply

There is no way to produce a pure GeO2 glass at the synthesis conditions we used in the research. Table 3 shows the densities and refractive indices of the glasses under investigation.

Comment #6

Excitation spectra of Fig. 8 should be shown.

Reply

Excitation spectrum fig. 8 shown in supplementary files (Fig. S19)

Comment #7

The PL peak deconvolution (Fig.9) is just a presentation. Add discussion about the relationship between the side peak and chemical compositions.

Reply

We attributed the deconvolution peaks with different BACs type in the Discussion section.

At the same time, calculations [11] showed that BACs were not individual low-valence bismuth ions, but complex systems of cations and an oxygen vacancy. In this case, both bands at 1125 nm (Peak 4 in Fig.9) and 1310 nm (Peak 3 in Fig.9) belonged to oxygen-deficient centers =Bi···Ge≡, and the difference in the bands position was caused by the presence or absence of aluminum ions in the second coordination sphere of the BACs, respectively [11]. It is known that luminescence in the longer wavelength part of the spectrum (1800–3000 nm) was observed in samples with a high content of Bi2O3 (>20 mol%), which was associated with the formation of Bi53+ cluster centers [28] or oxygen-deficient centers =Bi···Bi = [11]. We believe that luminescent BACs of two types were formed in our glasses during synthesis: namely, =Bi···Ge≡ (~1125, ~1310 nm) (Peak 4, 3 in Fig.9) and =Bi···Bi= (1615, 1885 nm) (Peak 2, 1 in Fig.9) in a smaller amount. Bi2O3 content growth led to an increasing of the =Bi···Bi= type centers number and to PL decreasing in ~1300 nm region. 

Comment #8

What does crystalline phase mean in Figure 11? The heat treatment gives the impression of a precipitation of single phase, even though no single phase has appeared in several glass-ceramics. It is confusing and should be removed.

Reply

We rearrange Fig.11 for clearence.

Figure 11 shows the characteristic temperatures of the glasses under investigation, the crystallization temperatures were attributed to the crystal phase formed during the heat treatment.